# Weighted Gene Co-Expression Network Analysis and Support Vector Machine Learning in the Proteomic Profiling of Cerebrospinal Fluid from Extraventricular Drainage in Child Medulloblastoma

**DOI:** 10.3390/metabo12080724

**Published:** 2022-08-05

**Authors:** Maurizio Bruschi, Xhuliana Kajana, Andrea Petretto, Martina Bartolucci, Marco Pavanello, Gian Marco Ghiggeri, Isabella Panfoli, Giovanni Candiano

**Affiliations:** 1Laboratory of Molecular Nephrology, IRCCS Istituto Giannina Gaslini, 16147 Genoa, Italy; 2Core Facilities–Clinical Proteomics and Metabolomics, IRCCS Istituto Giannina Gaslini, 16147 Genoa, Italy; 3Department of Neurosurgery, IRCCS Istituto Giannina Gaslini, 16147 Genoa, Italy; 4Division of Nephrology, Dialysis, Transplantation, IRCCS Istituto Giannina Gaslini, 16147 Genoa, Italy; 5Dipartimento di Farmacia (DIFAR), Università di Genova, 16147 Genoa, Italy

**Keywords:** medulloblastoma, brain tumor, artificial intelligence, mass spectrometry, extraventricular drainage, cerebral spinal fluid, proteomics

## Abstract

Medulloblastoma (MB) is the most common pediatric malignant central nervous system tumor. Overall survival in MB depends on treatment tuning. There is aneed for biomarkers of residual disease and recurrence. We analyzed the proteome of waste cerebrospinal fluid (CSF) from extraventricular drainage (EVD) from six children bearing various subtypes of MB and six controls needing EVD insertion for unrelated causes. Samples included total CSF, microvesicles, exosomes, and proteins captured by combinatorial peptide ligand library (CPLL). Liquid chromatography-coupled tandem mass spectrometry proteomics identified 3560 proteins in CSF from control and MB patients, 2412 (67.7%) of which were overlapping, and 346 (9.7%) and 805 (22.6%) were exclusive. Multidimensional scaling analysis discriminated samples. The weighted gene co-expression network analysis (WGCNA) identified those modules functionally associated with the samples. A ranked core of 192 proteins allowed distinguishing between control and MB samples. Machine learning highlighted long-chain fatty acid transport protein 4 (SLC27A4) and laminin B-type (LMNB1) as proteins that maximized the discrimination between control and MB samples. Machine learning WGCNA and support vector machine learning were able to distinguish between MB versus non-tumor/hemorrhagic controls. The two potential protein biomarkers for the discrimination between control and MB may guide therapy and predict recurrences, improving the MB patients’ quality of life.

## 1. Introduction

Brain tumors are the most frequent pediatric solid malignancies [1,2]. Medulloblastoma (MB) is the most common one, accounting for 20% of cases [1,2]. MB is a rare malignant embryonal brain tumor, classified into four principal molecular groups according to the 2021 WHO classification of tumors of the Central Nervous System (CNS), namely, wingless-activated (WNT), sonic hedgehog (SHH)-activated, and the non-SHH/WNT Group 3 and Group 4 [2]. In children, MB mostly arises in the cerebellum [1] with peak incidence around five years of age and a male prevalence [3]. Symptoms include headache, nausea, ataxia, and neuromotor impairments [4]. Diagnosis of MB relies on brain computed tomography followed by magnetic resonance imaging, although histopathology is required [4,5]. The current standard therapy includes surgery, followed by craniospinal irradiation and adjuvant chemotherapy [3]. Both pre-and intra-operative neuronavigation is revolutionizing pediatric neuro-oncology [6]. The development of high-throughput proteomic and genomic techniques has sensibly improved our knowledge of MB, an heterogeneous tumor, opening the way to the development of targeted therapy to improve overall survival rates [4,7,8,9]. Nonetheless, despite the advances in the molecular classification of pediatric brain tumors, there is the need to identify biomarkers for patient stratification, therapy response, and residual disease or recurrence detection [10,11]. Cerebrospinal fluid (CSF) is a promising source for biomarker discovery [12]. It is produced by the choroid plexus and originates in part from the CNS interstitial fluid [13]. CSF is in contact with the brain tissue (hence the tumor) and has been shown to contain many unique proteins [12,14,15,16,17]. A mass spectrometry (MS) comprehensive characterization of healthy CSF from lumbar puncture identified 2630 proteins [15]. Several proteomic studies of CSF from children with different types of brain tumors have been conducted [17,18,19]. The proteome of childhood CSF MB was analyzed by two-dimensional gel electrophoresis (2-DE) and MS, and prostaglandin D2 synthase was found to be downregulated in CSF from MB patients compared to controls [16]. The first comprehensive proteomic analysis of CSF from children with diffuse intrinsic pontine glioma found upregulation of Cyclophillin A and dimethylarginase [20]. Six potential biomarkers of metastatic spread were selected in a study of the CSF proteome of children bearing CNS tumors [17]. In the cited studies, CSF was derived either from lumbar puncture or intraoperative procedures [19]. By contrast, our previous proteomic study on different types of child brain tumors utilized CSF from extra ventricular drainage (EVD), identifying two biomarkers (TAF15 and S100B) able to discriminate between tumoral and non-tumoral conditions [21]. CSF proteins from children with glioblastoma were also studied by ELISA, and it was found that increased macrophage inhibitory cytokine-1/growth differentiation factor 15 (MIC-1/GDF15) predicted a shorter survival [22]. EVs are lipid-bound nanoparticles secreted by most cell types, as a mechanism of intercellular communication, present in all bodily fluids. EVs carry proteins, DNA, RNA, and lipids to nearby and distant cells [23]. Based on their size and biogenesis, EVs have been distinguished in exosomes (Ex, 40–120 nm), generated within the endosomal system, and microvesicles (Mv, 100–1000 nm) budding at the plasma membrane [23]. According to the guidelines of the International Society for Extracellular Vesicles of 2018 (MISEV2018) [24], the collective term “EV” is now recommended. Cancer-derived EVs were shown to affect the tumor microenvironment by remodeling the extracellular matrix, promoting tumor progression [25]. Therefore, EVs have potential as a tumor biomarker source [26]. In particular, Ex contribute to signal transduction between the tumor and the rest of the body, being involved in metastasis and therapy resistance [27]. By contrast, Mv resemble more closely the parental cell, being useful in assessing its oxidative and metabolic state [21]. In this respect, EVs represent promising new tools [27] in precision medicine for cancer liquid biopsy [26]. In this study, we sampled the proteome of waste CSF from extraventricular drainage (EVD) from six children with MB and six controls needing EVD insertion for unrelated causes. Total CSF, as well as its Mv, Ex, and proteins captured by peptide library beads (CPLL), was utilized. Findings provide new insight into MB pathogenesis, identifying long-chain fatty acid transport protein 4 (SLC27A4) and laminin B-type (LMNB1) and as candidate diagnostic biomarkers for MB.

## 2. Results

### 2.1. Characterization of Extracellular Vesicles

The size and homogeneity of Mv and Ex were confirmed by dynamic light scattering (DLS), revealing a Gaussian distribution profile with a typical mean peak at 1000 ± 5 or 100 ± 5 nm, respectively (Appendix A).

### 2.2. Protein Composition

We analyzed the proteome of the CSF from the EVD of six children bearing MB and six controls, fractionating it by four different biochemical–physical methodologies, obtaining, in addition to the total sample, its Mv and Ex fractions, as well as the sample equalized with CPLL beads, a solid-phase combinatorial library of hexapeptides coupled to poly(hydroxymethacrylate) beads, that reduced the protein dynamic range and enriched faint proteins (Figure 1).

In total, 3560 proteins were identified, of which 2412 (67.7%) were expressed in both groups. Only 346 (9.7%) and 805 (22.6%) proteins were exclusive to the control or MB samples, respectively (Appendix A). As far as the four samples are concerned, we identified 1329, 1214, 2061, and 1360 proteins in the total, CPLL, Mv, and Ex fractions, respectively, from controls, of which 584 (21.2%) were present in all fractions. Only 224 (8.1%), 167 (6.1%), 596 (21.6%), and 116 (4.2%) proteins were exclusive to the total, CPLL, Mv and Ex fractions, respectively (Appendix A). By contrast, in MB we identified 1220, 1533, 2397, and 2101 proteins in the total, CPLL, Mv, and Ex fractions, respectively, of which 701 (21.8%) were present in all fractions. Only 176 (5.5%), 232 (7.2%), 445 (13.8%), and 197 (6.1%) proteins were exclusive to the total, CPLL, Mv and Ex fractions, respectively (Appendix A). Interestingly, out of 1789 enriched proteins identified in the CPLL fractions from both groups, 1337 (75%) should be considered to be at a low or an extremely low concentration. This suggests that CPLLs were able to enrich the low-prevalence proteins. In fact, the concentration of 705 (39%) and 632 (35%) proteins, out of the cited 1337, was below the first and second quartile, respectively, of total protein detected in whole serum both with MS and immunoassay methods [28]. Further, out of 2539 proteins identified in the Mv fraction, 2371 (93%) have already been described in the Vesiclepedia database [29]. On the other hand, out of 2297 proteins identified in the Ex fraction, 1746 (76%) have already been described in the Exocarta database [30].

Despite considerable overlapping protein identity between the two clinical groups in the four fractions, multidimensional scaling analysis evidenced clear discrimination of three clusters consisting of the soluble proteins (total and CPLL fractions), Mv, and Ex, and, in each fraction, there was good discrimination between the control and MB samples (Figure 2).

To explore potential associations between control or MB, their fractions (total, CPLLs, Mv and Ex) and the clinical features, we utilized weighted gene co-expression network analysis (WGCNA) (Figure 1). WGCNA clusters identified proteins into modules of co-expression profiles that can be considered to be in a functional relationship with each other [31]. In this way, WGCNA identified which module and protein expression profile was functionally associated with the total, CPLL, Mv and Ex fractions from either the control or MB condition. This analysis revealed nine modules encompassing proteins with similar co-expression profiles. To distinguish between modules, an arbitrary color was chosen for each one (Appendix A). The number of proteins included in each module ranged from 29 (white module) to 1276 (green module). The red and white modules showed statistically significant relationships with MB (r = 0.8) and control (r = 0.7) samples, respectively. In particular, the white module was enriched in proteins involved in the regulation of immune response and cell migration, whereas the red one was enriched in proteins involved in the binding of proteins, nucleotide phosphorylation, regulation of the immune system, signaling by interleukins, regulation of metabolism, glycolysis, and gluconeogenesis. No other statistical relationship was found between modules and other clinical traits, including age and gender. The Spearman correlation coefficient of all proteins with each trait is reported in Appendix A, and their profile is visualized by a heatmap diagram (Appendix A). In the heatmap, each row represents a protein, and each column corresponds to a trait. Normalized Z-scores of the protein Spearman correlation coefficients are depicted by a pseudocolor scale, with red, white, and blue indicating a positive correlation, no correlation, and a negative correlation, respectively. The tree dendrogram displays the results of unsupervised hierarchical clustering analysis, placing similar Spearman correlation coefficient values next to each other. At-test was applied to identify the proteins that distinguished between the control and MB in each fraction. Out of 446 proteins identified, 68, 22, 163, and 224 were statistically significant in the comparison of the control and MB for the total, CPLL, Mv, and Ex fractions, respectively (Appendix A). Out of 446 statistically significant proteins, 62 have already been reported to be associated with MB [32]. PLS-DA and SVM learning analyses were performed to reduce the number of selected proteins that maximized the discrimination between the control and MB groups in the four fractions and to prioritize their importance (Figure 1). These analyses identified a ranked core panel of 192 proteins, among which 31, 8, 71, and 89 were highlighted (Appendix A). Their priority was determined using the rank list and the variable importance in projection (VIP) score obtained using SVM and PLS-DA, respectively. Both analyses identified the same protein priority order. The expression profile of this core panel of proteins, after Z-score normalization, is visualized in the heatmap shown in Figure 3.

The k-means analysis associated with PLS-DA showed the presence of eight distinct clusters, corresponding to the four samples of the two clinical groups (Figure 4).

Among the 192 highlighted proteins, SLC27A4 and LMNB1 were the most promising potential biomarkers to distinguish the control from MB samples in the four fractions tested. Indeed, in the whole core panel of proteins, in all the *t*-test comparisons, these two proteins displayed high *p* values and maximal discrimination power in SVM and PLS-DA. In particular, SLC27A4 was associated with controls, whereas LMNB1 was associated with MB samples. The considerable diversity in the expression profile of the proteins identified in the control and MB samples in the four fractions may imply their different physiological roles. To assess this, we performed Gene Ontology (GO) enrichment analysis based on the annotation terms extracted from the Gene Ontology Consortium (http://www.geneontology.org/ accessed on 16 February 2022). This analysis identified 60 significantly enriched GO annotation terms. Among these, 32 and 28 were enriched in MB or controls, respectively (Appendix A). Results are visualized by a scatter plot (Appendix A), where each point corresponds to a GO annotation term. The points above or below the line with equation x = y were enriched in the respective MB or control samples. Interestingly, control samples were enriched in proteins involved in cell adhesion and in the regulation of immune response, while MB samples were enriched in extracellular vesicle and nucleus proteins and in proteins involved in mRNA processing.

### 2.3. Western Blot for LMNB1

Western blots were used to confirm the presence of LMNB1 in the total fraction of CSF from the EVD of the control and brain tumor patients. As shown in Figure 5a, a monoclonal anti-human LMNB1 detected a single band corresponding to the molecular weight of LMNB1 (66 kDa) [32]. This band was much more abundant in CSF from the EVD of MB compared to controls.

### 2.4. ELISA for LMNB1-Validated Proteomic Results

ELISA was used to validate the proteomic results regarding LMNB1 expression in the total fraction of CSF from the EVD of the control, MB, and other brain tumors (Figure 1). Figure 5b shows that LMNB1 was statistically more abundant (*p* < 0.001) in CSF from the EVD of MB compared to the control and to LGGs and GN, and other brain tumors, namely, 1.4 (1.1–1.9), 0.46 (0.28–0.5), 0.5 (0.3–0.66), and 0.56 (0.5–0.64) Relative Units per milliliter (RU/mL), respectively (Figure 5c). The comparison of MB with all other samples is displayed in the area under the curve (AUC), confidence interval (CI), and *p* value of 0.97 (0.91–1) and *p* < 0.001, respectively (Figure 5d). Finally, the cutoff, sensitivity and specificity percent, and likelihood ratio of the assay were 0.74 RU/mL, 92 (61–99.8) (%), 96 (86–99.5) (%), and 22.5, respectively.

## 3. Discussion

In this study, we present a comprehensive analysis, conducted by liquid chromatography-coupled tandem MS, of the proteome of human CSF from EVD and of the Mv and Exo, as well as proteins isolated using CPLLs. The novel aspects of our study, besides the analysis of multiple CSF fractions, are the sampling of CSF from EVD, which allowed us to overcome the ethical issues related to its collection, and the sample low abundance when drawn by lumbar puncture. Notably, the World Health Organization has classified MB as a grade IV tumor, originating in the cerebellum and spreading via the CSF to the brain, which renders this sample central in the study of the MB biology and dissemination [1]. We have examined the cited four fractions isolated from human EVD CSF in order to implement the number and type of identified proteins. Accordingly, out of a total of 3560 proteins identified in the CSF from control and MB patients, we were able to identify and discriminate the single components of the CSF from EVD. Notably, notwithstanding the protein identity overlap between the four fractions, MDS analysis highlighted three clusters for each clinical condition, encompassing the soluble proteins (total plus the CPLL fraction), the Mv, and Ex fractions, with good discrimination between the control and MB samples in each fraction. In this setting, we made novel observations, such as the fact that the most discriminating proteins were enriched in the Ex fraction and that the faint proteins were enriched in the CPLL fractions. The latter can in fact reduce the protein dynamic range. We were also able to confirm one of the newest and unique features of the Exo, namely, their enrichment in proteins involved in oxidative phosphorylation, the Krebs cycle, and pathways, a feature probably transversal in these vesicles, conferring them the ability to remain viable for a long time [34]. We also observed that, while Ex play a signaling role, Mv have the ability to recapitulate more closely the metabolic and redox status of the donor cells, as they derive from the plasma membrane [35]. Consistently, here we found that Mv from MB CSF are enriched in the enzymes of glucose and fatty acid metabolism with respect to Mv from control CSF (Appendix A). Tumor-secreted EVs are important mediators of intercellular communication in the local tumor microenvironment (TME) [36]. Proteomics of Ex from both an MB cell line and patient serum suggested that the properties of the Ex are relevant to MB biology [37]. A significant increase in EV release was reported in EGFRvIII-expressing glioma cells compared to EGFRvIII-negative ones [38]. Notably, EVs have been shown to be able to traverse the blood–brain barrier [39]. The use of EVs is promising as these can be isolated from the plasma of patients and therefore may in principle be useful for non-invasive diagnosis of brain tumors. For example, EVs isolated from the plasma of patients with high-grade gliomas, analyzed with the tandem mass tag labeling LC-MS/MS method, showed high levels of syndecan-1 (SDC1), which can discriminate between high-grade glioblastoma multiforme and low-grade glioma [40]. The WGCNA constructed nine co-expression modules, with only the red and white ones showing a high correlation with MB and control samples, respectively. Bioinformatics showed a ranked core list of 192 proteins that allowed distinguishing between the control and MB. Among these, artificial intelligence highlighted SLC27A4 and LMNB1 as proteins that maximize the discrimination between cancer and non-cancer samples, with SLC27A4 associated with controls, and LMNB1 with MB samples. ELISA analysis of total CSF validated this observation. Moreover, ROC curve analysis indicated that LMNB1 has good diagnostic value as a biomarker for MB. LNMB1 is a nuclear structural protein member of the lamin protein family [41], involved in DNA replication, transcription, and repair, nuclear autophagy, and cytoskeletal interactions and is considered a marker of cellular senescence [42,43]. LMNB1 has been identified in several studies searching for tumor biomarkers, many of which have demonstrated that it plays a vital role in carcinogenesi [44]. A study analyzing differently expressed genes of the 7 Gene Expression Omnibus (GEO) database by the Robust Rank Aggregation (RRA) method proposed LMNB1 as a potential biomarker for glioma, as patients with high expression of LMNB1 had lower survival rates, while silencing of LMNB1 inhibited the proliferation of glioma cells [45]. LMNB1 might play a crucial role in carcinogenesis in MB. Consistently, a proteogenomic characterization of pediatric MB tissue samples conducted by quantitative proteomics investigation using SILAC found that LMNB1 was enriched in the MB samples relative to cerebellum controls (fold-change of 6.99) [42]. Microarray datasets from the Gene Expression Omnibus (GEO) specimens showed that the LMNB1 gene was upregulated in prostate cancer samples, in a manner associated with a higher tumor grade [43]. Genotyping of potentially clinically relevant single-nucleotide polymorphisms (SNPs) in childhood acute lymphoblastic leukemia patients identified variants in the *LMNB1* gene as related to the risk of relapse [46]. The role of LMNB1 appears to be different in the various types of cancer. Higher LMNB1 mRNA expression in lung cancer cells, with respect to normal tissues, was related to tumor stage and adverse overall survival, while LMNB1 knock down inhibited lung cancer cell proliferation [47]. LMNB1 was up-regulated in a 5-FU-resistant HCT116 human colon cancer sample cell line [48]. By contrast, LMNB1 was among the top25 down-regulated proteins in quantitative stable isotope labeling using amino acids in cell culture (SILAC) proteomic analysis of human colon cancer cells induced by galectin-4 [49]. A study of LMNB1 mRNA expression in breast cancer tissues found that lower LMNB1 transcript levels were associated with worse clinical outcome [50]. Proteomic-wide profiling of the plasma of hepatocellular carcinoma (HCC) patients found that LMNB1 was significantly upregulated and associated with tumor size and stage [51]. SLC27A4 belongs to the Solute-carrier family 27A molecules (SCL27As) of fatty acid transport proteins, involved in the translocation of long-chain (>12 C) fatty acids across the plasma membrane [52].The human SLC27A family, also known as fatty acid transport proteins (FATPs), includes six members (SLC27A1/Slc27a1 to SLC27A6/Slc27a6) that play an important role in lipid metabolism. Among these, SLC27A4/Slc27a4 (FATP4/Fatp4) is expressed in the brain, kidney, liver, intestine, skin, heart, and endothelial cells [52]. The SCL27As were shown to play a crucial role in multiple malignant tumors, regulating long-chain fatty acid uptake and lipid metabolism. SLC27A4 is related to very long chain acyl-CoA synthetase activity. It was upregulated in HCC cell lines with respect to normal hepatocyte cell lines, while its downregulation suppressed the invasivity of those cells [53]. SLC27A4 was also overexpressed in lung tumor tissues [54] and in triple negative breast cell lines (Hs578T and MDA-MB-231) with respect to non-tumor tissues [55]. However, the expression of SCL27As plays different roles in the progression of the disease [56]. The present data, showing that SLC27A4 is downregulated in MB samples, suggest that downregulation of SLC27A4 alters the uptake of specific fatty acids and their utilization by MB cells. SLC27A4 knockout was shown to significantly reduce linoleic acid movement [57]. On the other hand, oleic acid (C18:1), which has antitumor effects in several types of cancers [58], is a preferred substrate of SLC27A1 and SLC27A6, but not SLC27A4 [59]. Although the brain relies essentially on glucose, lipid metabolism is crucial for this lipid-rich organ, where polyunsaturated fatty acids are highly represented, playing structural and signaling roles [60]. Due to the presence of the blood–brain barrier, fatty acid transport proteins are involved in lipid transfer from circulation. Based on human expression studies, SLC27A1 and SLC27A4 are the predominant fatty acid transport proteins expressed in the blood–brain barrier [57]. Metabolic reprogramming is a hallmark of cancer [61]. For example, it has been shown that Serine–glycine-one-carbon (SGOC) metabolism is a considerable part of neuroblastoma metabolism [62]. A metabolic feature of cancer cells is their preference for aerobic glycolysis (also called the Warburg effect [63]), which has recently been reinterpreted as a process supporting the anabolic pathways needed for proliferation at the expense of less efficient ATP production [64]. On the other hand, increased mitochondrial mass and oxidative phosphorylation benefit at least some aspects of metastasis [65]. Considering that the present study utilized CSF samples from a variety of MB subtypes, which may rest at various stages of development, it may be presumed that metabolic reprogramming occurs in the MB subtypes that we are witnessing altogether. The GO enrichment analysis showed that controls were enriched in processes related to cell adhesion and regulation of the immune response, which is conceivable as the loss of cell-to-cell controls is typical of neoplasms, while MB samples were enriched in proteins involved in mRNA processing and nucleus and EV proteins. Notably, EVs such as Ex can mediate intercellular communication between TME components [65]. EVs also mediate aspects of communication between tumor cells and other cells of the TME. Mitochondrial metabolites and proteins are critical cargo of EVs [36,66]. In the present study, Ex expressed the main proteins of the respiratory chain, F_1_F_o_-ATP synthase, and Krebs Cycle (Appendix A). Cancer cells were reported to internalize Ex, whose cargo significantly altered the target cancer cell metabolism [29]. Interestingly, MB cells were shown to undergo metabolic reprogramming, dependent on EV-based communication of cells within the TME: a gene expression signature of EVs from MB patients, encompassing LDHA (encoding subunit A of lactate dehydrogenase), a protein involved in anaerobic glycolysis, was upregulated in Group 3 MB subtype [67]. The subunit A of lactate dehydrogenase (LDHA) was also overexpressed in our MB samples, being found in the red module, related to the MB condition. LDH activity may play a role in the acidification of the TME. The TME of the SHH-MB subgroup was studied by mosaic analysis with double markers, and it was found to contain astrocytes active in secreting IL4, in turn stimulating IGF1 production by tumor-associated microglia, whose metabolic signalling was involved in MB progression [68]. Key prognosis questions in MB depend on treatment tuning to maximize overall survival. The standard of care for MB, consisting of CSI and adjuvant chemotherapy after surgical resection, can cause side-effects. The need for new tailored therapeutic approaches requires the identification of specific biomarkers [4]. Early diagnosis can also ameliorate the patient’s outcome, as metastases are observed in 30% of MB patients at diagnosis [69]. Quantitative proteomics, with its systems biology translational potential, along with AI use allowed us to study the complexity of MB cancer biology, and the identification of novel biomarkers has the potential to pave the way for the design of targeted interventions.

## 4. Materials and Methods

### 4.1. Sample Collection and Patient Information

A total of 61 pediatric patients with a brain tumor or with congenital hydrocephalus (grades III to V), admitted to the Neurosurgery Unit of Istituto Giannina Gaslini Children’s Hospital in the period 2019–2022, who required placement of an EVD catheter, were included in the study. All patients treated for a brain tumor had histological diagnosis centrally reviewed and performed according to the World Health Organization (WHO) classification [66]. CSF samples otherwise destined for waste were collected non-invasively at the first change of the disposable bag of EVD by a sterile procedure (no seriate sampling was performed). Written informed parental consent was obtained before enrolment. The main demographic and clinical features of the subjects are summarized in Table 1.

Twelve randomly selected patients from MB (six) and control (six) groups were included in the proteomic analysis. A group of 61 CSF samples including those from the cited 12 subjects, plus samples from 18 congenital hydrocephalus unrelated to a brain tumor, 16 low-grade gliomas and glioneuraltumors, 6 embryonal tumors, and 9 other brain tumors was included in the validation part of the study. All EVD CSF samples were centrifuged at 3000× *g* for 10 min to remove cells and debris, aliquoted in different vials, and immediately frozen at −80 °C until use. Samples to be analyzed by proteomics analysis were fractionated by means of four different biochemical–physical methodologies, obtaining, besides the total sample, its Mv and Ex fractions, and the sample equalized with CPLL beads, a solid-phase combinatorial library of hexapeptides coupled to poly(hydroxymethacrylate) beads, which reduce the protein dynamic range and enrich faint proteins, so to increase the type and number of identifiable proteins. The sample size for each experiment was determined according to Dell et al. [70]. The study was carried out in accordance with Italian and international ethical guidelines and was approved by the local Ethics Committee (n. 18 of 31 October 2013, protocol n. 176, Ethic committee of “G. d’Annunzio” University and ASL N.2 Lanciano-Vasto-Chieti, Italy).

### 4.2. Total Fraction

The total fraction was obtained from aliquots of 1 mL of CSF EVD as reported by Bruschi et al. [21].

### 4.3. CPLLs Fraction

The CPPL fraction was obtained from aliquots of 50 mL of CSF EVD as reported by Santucci et al. [71].

### 4.4. Extracellular Vesicle Fraction

Mv and Ex were isolated from aliquots of 50 mL of CSF EVD by centrifugation as reported by Bruschi et al. [72].

### 4.5. Dynamic Light Scattering

Mv and Ex size and homogeneity were determined by dynamic light scattering (DLS) as reported by Bruschi et al. [72].

### 4.6. Mass Spectrometry (MS) Analysis

All samples were treated as reported in Bruschi et al. [73]. Briefly, samples were lyzed, reduced, and alkylated with iST-LYSE buffer (PreOmics, Klopferspitz, Germany), digested, and processed by the iST protocol [74]. Digested samples were eluted with a 200 cm µPAC C18 column (ThermoFisher Scientific, Waltham, MA, USA), and the peptides were separated using a non-linear gradient of 5–45% solution of 80% *v*/*v* acetonitrile, 5% *v*/*v* dimethyl sulfoxide, and 0.1% *v*/*v* formic acid in 155 minutes at a flow rate of 350 nL/min. MS data were acquired on an Orbitrap Fusion Tribrid mass spectrometer (ThermoFisher Scientific, Waltham, MA, USA). Raw data were processed with MaxQuant [75] software version 1.6.10.0. A false discovery rate (FDR) of 0.01 was set for the identification of proteins, peptides, and PSM (peptide-spectrum match). For peptide identification, a minimum length of 6 amino acids was required. Andromeda engine, incorporated into MaxQuant software, was used to search MS/MS spectra against the Uniprot human database (release UP000005640_9606 April 2019). In the processing, acetyl (protein N-term), oxidation (M), and deamidation (NQ) were selected as variable modifications, and the fixed modification was carbamidomethyl (C).

### 4.7. Western Blotting

One ml of CSF from the EVD of control or MB pooled samples was separated by sodium dodecyl sulfate polyacrylamide gel electrophoresis (SDS-PAGE) and then transferred to a nitrocellulose membrane. The full-length membrane was blocked, rinsed, labeled, and detected with monoclonal anti-human LMNB1 diluted in 3% *w*/*v* bovine serum albumin (BSA) in PBS containing 0.05% *v*/*v* Tween-20 (PBS-T). After rinsing in PBS-T, the membrane was incubated with HRP-conjugated secondary antibodies (diluted 1:10,000 in 1% *w*/*v* BSA in PBS-T). The chemiluminescence signal was acquired and quantified using ChemiDoc and Quantity One software (Bio-Rad, Hercules, CA, USA).

### 4.8. ELISA

To confirm the results obtained by high-resolution MS for LMNB1 expression in the EVD CSF of control and MB patients, homemade ELISA was performed on a group of 61 patients consisting of the previous 12 subjects with the addition of 18 congenital hydrocephalus samples unrelated to a brain tumor, 16 low-grade gliomas and glioneural tumors, 6 medulloblastoma, and 9 other brain tumor samples. This immunoassay utilized the technique of a direct ELISA. Briefly, one ml of CSF EVD sample was precipitated as described in the total fraction section, and the pellet was solubilized in 100 µL of a solution consisting of PBS, 1% *v*/*v* NP40, 0.1% *w*/*v* sodium deoxycholate, 0.1% *w*/*v* SDS, 1 mM EDTA, and protease inhibitor cocktail. Then, the sample was added to the plate wells and incubated overnight at 4 °C. After one wash with PBS, we added 200 µL of 3% *w*/*v* BSA in PBS to block the remaining absorption sites, left overnight at 4 °C. After three washes with 0.05% *v*/*v* Tween-20 in PBS (PBS-T), we added 100 µL of monoclonal anti-human LMNB1 (1:2000) in 1% *w*/*v* BSA in PBS-T, left overnight at 4 °C. After three washes with PBS-T, we added 100 µL of secondary antibody HRP conjugated for 2 h at room temperature. After three washes, 100 µL of TMB substrate solution was added. The reaction was stopped with 100 µL of 1 M H_2_SO_4_, and the absorbance was read at 450 nm in an iMark microplate reader (Bio-Rad, Hercules, CA, USA). All washes were performed by gentle shaking. The assays were performed on 96-well Nunc MaxiSorp plates (ThermoFisher Scientific, Waltham, MA, USA). Box plots were used to visualize the protein levels. In the box plots, each point indicates a patient. The lowest detection limit was determined as the lowest protein amount that could be differentiated from the blank.

### 4.9. Statistical Analysis

MS data, after normalization, were analyzed by unsupervised hierarchical clustering using multidimensional scaling (MDS) with k-means and Spearman’s correlation to identify outliers and possible dissimilarity between samples. Then, a co-expression network was constructed with the normalized whole dataset using the weighted gene co-expression network analysis (WGCNA) package in R [31]. A weighted adjacency matrix was constructed using the power function. Once the appropriate power β parameter (independence scale value set at 0.8) was chosen, the adjacency matrix was transformed into a topological overlap matrix (TOM). TOM measures the network connectivity of all proteins. To place the proteins displaying co-expression profiles into protein modules, hierarchical clustering analysis was conducted according to the TOM dissimilarity, with 20 proteins per module minimum size. To study the relationship between modules and clinical traits, we used the module eigengenes (MEs) and calculated the Spearman’s correlation between MEs and the traits (namely, MB, control, total, CPLLs, Mv, and Ex). Proteins were considered to be correlated with one trait with a significant (two-sided *p* value ≤ 0.05 after Benjamini–Hochberg correction for multiple interactions) Spearman’s correlation coefficient value (>0.7). To identify the hub proteins in the modules that maximized the discrimination between MB and control samples for each trait, we applied a *t*-test and machine learning methods such as non-linear support vector machine (SVM) learning and partial least squares discriminant analysis (PLS-DA). For the *t*-test, proteins were considered to be significantly differentially expressed between two conditions with a power of 80% and an adjusted *p* value ≤ 0.05 after correction for multiple interactions (Benjamini–Hochberg) and a fold change of ≥2. Moreover, the proteins needed to show at least 70% identity in the samples in one of two conditions and the area under the curve (AUC) in the receiver operating characteristic (ROC) analysis >0.7. Volcano plots were used to visualize the expression fold-change differences between each comparison with the cutoff line being established using the function y = c/(x − x_0_). To estimate the prediction and classification accuracy of the SVM, kernel linear algorithm and a fourfold cross-validation approach were applied. A confusion matrix was used to visualize the results. The whole matrix was randomly divided into two parts: one for learning (65%) and the other (35%) to determine the prediction accuracy. The same algorithm was used to establish the ranked features in the classification. Gene set enrichment analysis [76] was utilized to build a protein functional network based on the Gene Ontology (GO) annotations extracted from the Gene Ontology Consortium (http://www.geneontology.org/, accessed on 16 February 2022). After loading the protein expression data in the dataset, a ranked list was assigned to each GO annotation/pathway. These ranks consider the number of proteins associated with each gene signature with respect to all proteins, their mean fold change, and the *p* value after correction with the False Discovery Rate (FDR) for multiple interactions. These rank values lie between −1 and 1, corresponding to minimal and maximal enrichment in each group. In the two-dimensional scatter plot, the points that belong to the line passing between the points with coordinates (1_x_,1_y_) and (−1_x_, −1_y_) represent the equally enriched signatures. The distance from this straight line (above or below the line indicate the GO annotation terms positively enriched in MB or control samples, respectively) is proportional to the increase in the signature enrichment in one of the two groups. To assess the difference in the levels of LMNB1 between control, MB, and other brain tumor samples in the ELISA, the Kruskal–Wallis test for unpaired samples with Dunn’s multiple comparison test correction was used. Results are expressed as medians and the interquartile range (IQr). To evaluate the diagnostic efficiency of the assay, receiver operating characteristic (ROC) curves were generated. AUC values were classified as follows: 0.5, not discriminant; 0.5–0.6, fail; 0.6–0.7, poor; 0.7–0.8, fair; 0.8–0.9, good, and 0.9–1, excellent. Youden’s index and likelihood ratio were used to identify the cutoff and the diagnostic performance, respectively, of each assay. Two-sided *p* values ≤ 0.05 were considered significant. All statistical tests were performed using Origin Lab V9, Graphpad Prism, and the latest version of the R software package available at the time of the experiments.

## Figures and Tables

**Figure 1 metabolites-12-00724-f001:**
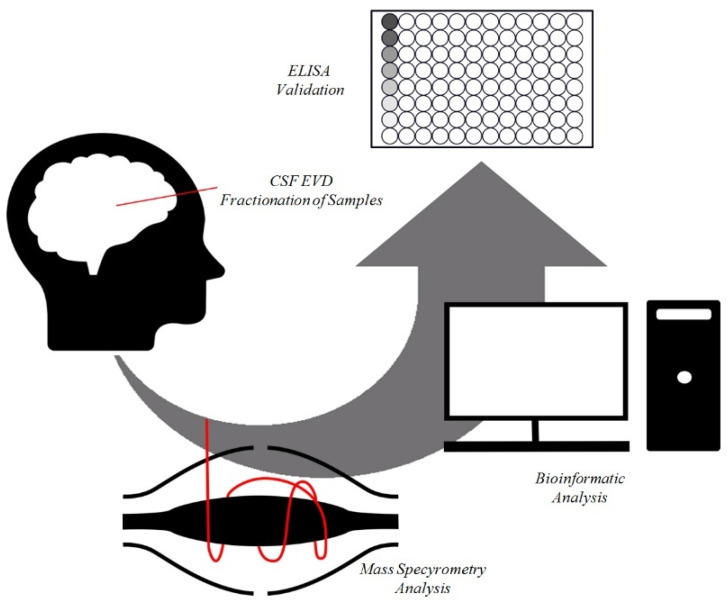
Workflow of the analysis. Cerebrospinal fluid from extraventricular drainage samples were collected, fractionated, and analyzed by mass spectrometry. The whole dataset was analyzed by the combined use of statistical and bioinformatic analyses to identify new potential biomarkers of medulloblastoma. The results of these analyses were validated by enzyme-linked immunosorbent assay (ELISA).

**Figure 2 metabolites-12-00724-f002:**
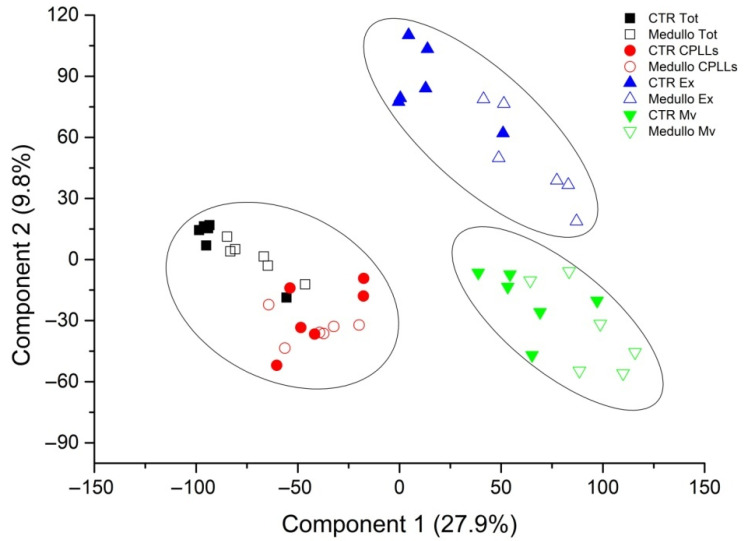
Multidimensional scaling (MDS) analysis of the proteins identified in CSF from EVD in MB or control samples. The two-dimensional scatter plot of MDS analysis shows the unsupervised cluster analysis of all CSF proteins identified from the EVD of MB (open symbol) and controls (solid symbol) in total (black square), CPLL (red circle), Mv (green triangle), and Ex (blue triangle) fractions. Ellipses show the 95% CI of the two clusters. MDS analysis identified three clusters corresponding to soluble proteins (total and CPLL fraction) and the extracellular vesicle fractions distinct in Mv and Ex fractions. No outliers were detected.

**Figure 3 metabolites-12-00724-f003:**
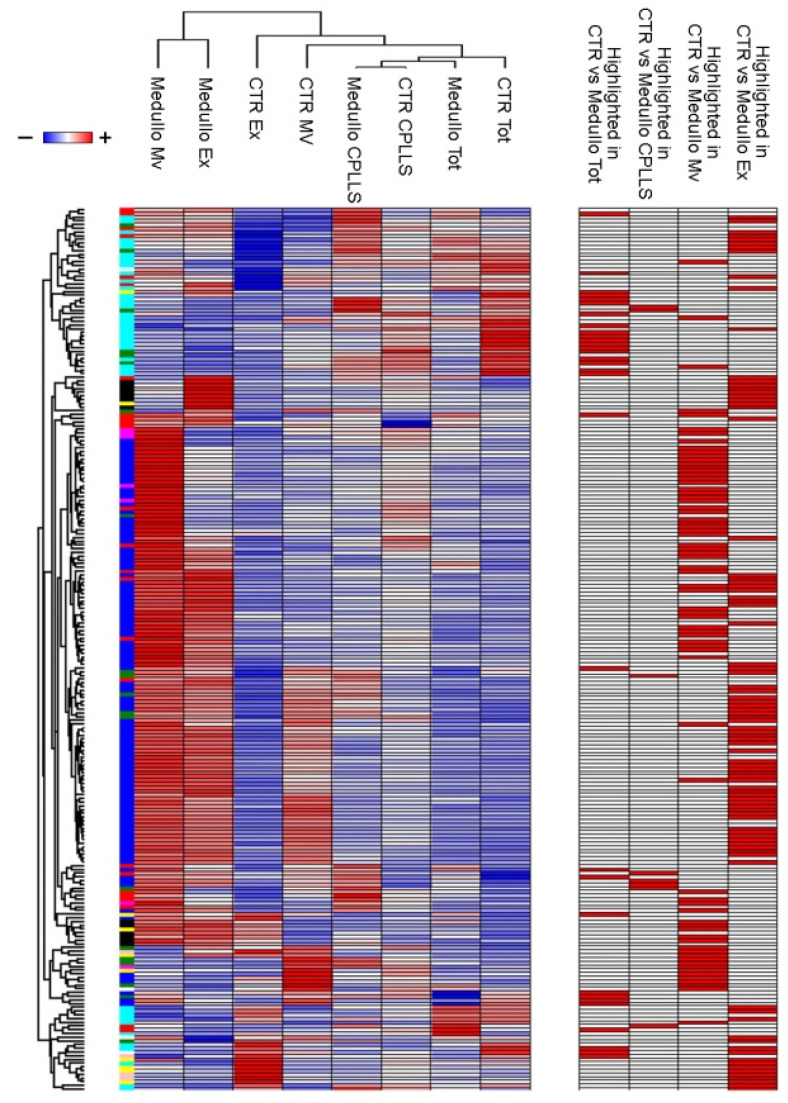
Heatmap of proteins that maximize the discrimination between MB and control samples. Heatmap of 192 proteins highlighted by the combined use of a *t*-test, Partial Least Square Discriminant Analysis (PLS-DA), and Support Vector Machine (SVM) learning analysis. In the heatmap, each row represents a protein, and each column corresponds to a sample (total, CPLL, Mv, and Ex of CSF from the EVD of MB and control patients). Normalized Z-scores of protein abundance are depicted by a pseudocolor scale (with red indicating positive expression; white, equal expression; and blue, negative expression, compared to each protein value). The dendrogram displays the outcome of unsupervised hierarchical clustering, placing similar proteome profile values next to each other. The co-expression modules identified are reported on the left of the heatmap. The statistically significant proteins identified in each comparison are highlighted in red on the right of the heatmap.

**Figure 4 metabolites-12-00724-f004:**
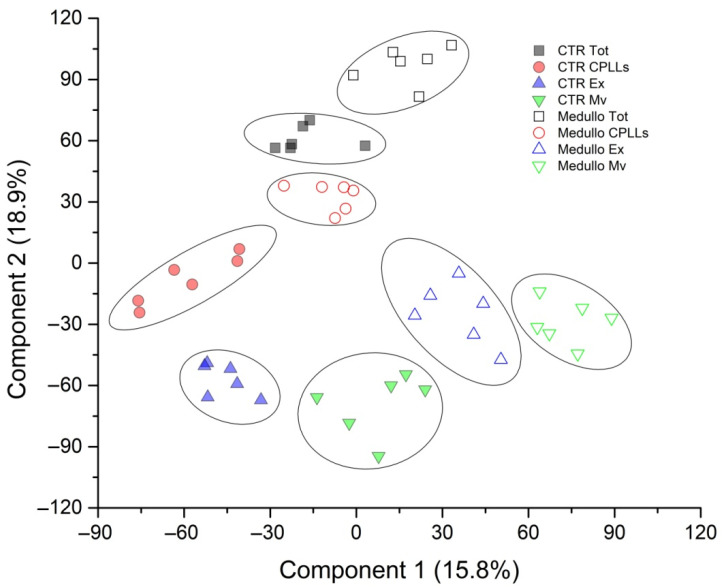
Partial Least Square Discriminant Analysis (PLS-DA) of all proteins identified in CSF from the EVD of MB and control samples. Two-dimensional scatter plot of PLS-DA showing the supervised cluster analysis of all proteins identified in CSF from the EVD of MB (open symbols) and control (solid symbols) in the total (black squares), CPLL (red circles), Mv, and Ex of the two clusters. PLS-DA identified eight clusters corresponding to the two clinical groups in each fraction.

**Figure 5 metabolites-12-00724-f005:**
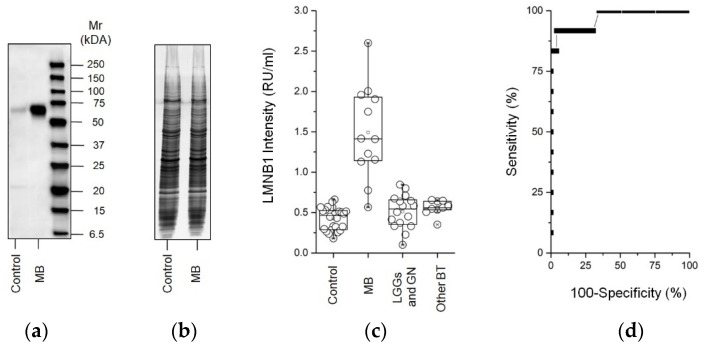
LMNB1 evaluation of the total fraction of CSF from the EVD of MB or control patients. (**a**) Representative western blot analysis and (**b**) gel stained with Blue Silver according to Candiano G. et al. [33] (used as a loading control) of full length gel (12–16T%) for LMNB1 in the total fraction of CSF from the EVD of control and Medulloblastoma (MB); (**c**) Box plot showing the median and interquartile range value of the ELISA of the LMNB1 protein in the total fraction of CSF from the EVD of the control, MB, low-grade gliomas and glioneural tumors (LGGs and GN), and nine other brain tumors patients. LMNB1 proteins were more abundant in MB compared to all other patients (*p* < 0.0001); (**d**) ROC curve analysis for the LMNB1 assay.

**Table 1 metabolites-12-00724-t001:** Clinical characteristics of the 61 patients in the study. All patients with brain tumors received a histological diagnosis. The abbreviations MS and ELISA indicate that samples were analyzed by MS and/or ELISA, respectively. The total patient number in each clinical group is shown in square brackets. Age is reported as the mean (years) and the range is shown in round brackets.

Groups	MS/ELISA	Sex (F/M)	Age (Year)
**Control [24]**			
Congenital hydrocephalus [24]	6/24	14/10	1(0–22)
**Low-grade Giomas and Glioneural Tumors [16]**			
Pilocytic astrocytoma [8]	0/12	6/6	8(3–15)
Gangliocytoma/Ganglioglioma [3]	0/4	2/2	9(5–11)
**Medulloblastoma [12]**	6/12	4/3	5(0–15)
**Other [9]**			
Meningiomas [2], Germ Cell Tumors [2], Epindimomas [2], Plessopapillomas [2], Emangioblastoma [1]	0/9	3/6	9(0–18)

## Data Availability

Whole-mass spectrometry data are available from the ProteomeXchange Consortium [77] via the PRIDE58 partner repository with the dataset identifier PXD035292.

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
