# Peer review of "Weighted Gene Co-Expression Network Analysis and Support Vector Machine Learning in the Proteomic Profiling of Cerebrospinal Fluid from Extraventricular Drainage in Child Medulloblastoma"

_metabolites, 2022, doi:10.3390/metabo12080724_

Round 1

Reviewer 1 Report

in this article, Bruschi et al, aimed to assess biomarkers of pediatric  Medulloblastoma via a proteomics approach from the CSF collected from Extraventricular Drainage in Child.

the work has scientific merit, however, it is written in such a way that requires extensive English editing where there are wrong typos (first, should be first), undefined acronyms (TMT), and wrong sentences (from others REF NS.).

Major Concerns

as to the study design, the write-up is so problematic, in the abstract they mention that they included 6 patients with different subtypes of MB.

This is a major error as we can not categorize these as one group.

then the other controls are with other disorders other than MB and they were considered as Control !!!!

the work has performed proteomics on EV, CSF, and MV; however, the data presented and validated is on the CSF; so what is the value of doing EV and MV and EV.

Validation of Western blot should be performed on the 6 samples and the raw data should be presented.

Validation of the other marker SLC27A$ should've been performed.

the title is misleading as WGCNA is not an artificial intelligent approach but a method to value the correlation among network analysis.

Pathway analysis and systems biology should have been performed

no power analysis was presented to the 6 samples and they are highly heterogeneous

Author Response

Open Review-1

Comments and Suggestions for Authors

in this article, Bruschi et al, aimed to assess biomarkers of pediatric  Medulloblastoma via a proteomics approach from the CSF collected from Extraventricular Drainage in Child.the work has scientific merit,

We are grateful to the Reviewer for the overall positive evaluation of our work.

however, it is written in such a way that requires extensive English editing where there are wrong typos (first, should be first), undefined acronyms (TMT), and wrong sentences (from others REF NS.).

We apologize for the grammatical and typographical errors; the English Language was corrected throughout, as were the undefined acronyms. Indeed, TMT stood for tandem mass tag labelling, a technique utilized by the Authors of ref Nr.42 (doi: 10.1158/1078-0432.ccr-18-2946.), the abbreviation was eliminated in the revised version of the manuscript. The other abbreviations have been checked and added, as the paragraph “Abbreviations” at the end of the text.

Major Concerns

as to the study design, the write-up is so problematic, in the abstract they mention that they included 6 patients with different subtypes of MB.This is a major error as we can not categorize these as one group. then the other controls are with other disorders other than MB and they were considered as Control !!!!

Indeed, we do not think that the inclusion of 6 patients with different subtypes of MB can be considered a major error; conversely it was done on purpose, to identify all-encompassing diagnostic biomarkers of MB, regardless of their subtype. As for the number itself, 6 were the children bearing MB in the period in our IRCCS. Nonetheless the numerosity allowed to draw statistical conclusions. The aim of our study was to make use of the unique opportunity offered by the considerable volume of “waste” CSF from EVD for proteomic analysis in seek for diagnostic protein biomarkers of MB, compared to a number of (non-physiological) non-tumor related conditions, which renders our controls more correct, as the EVD insertion could per se be considered otherwise confounding. As specified above, our controls were intentionally CSF samples from all patient that needed an EVD insertion for any causes unrelated to tumor. In this respect, an haemorrhagic and inflammatory condition would be an inclusion criteria, for the reasons above specified. Any other condition needing an EVD, although characterized by inflammation and pathological CSF production (our controls), if we had taken CSF from lumbar puncture as control, which also would never be "normal" for ethical issues. 

Also, most studies collected CSF by lumbar puncture or surgery around the brain area, this would suffer from the bias of a sampling site far from local interactions of the tumor with the brain tissue which may hamper the possibility to find specific proteins that may locally leak from the tumoral cells. In this respect we felt that the overproduced hence abundant CSF would allow to unveil proteins that may not be found in other ways. This study intended to pave the way to overcoming the difficulties encountered in assessing response to therapy, residual disease and recurrence in tumour management, by identifying biomarkers that, once validated, could be searched for also in other body fluids.

the work has performed proteomics on EV, CSF, and MV; however, the data presented and validated is on the CSF; so what is the value of doing EV and MV and EV.

the advantage of fractionating the sample into its various components is that it thus increases the probability of finding diagnostic biomarkers.

the samples are homogeneous i.e., they are all MB, although of different subtypes, instead they are heterogeneous in the controls, which allows us to see a large case mix.

In the validation, we used glioblastoma and other brain tumor types for the ELISA, which were purposely chosen because they were different from each other just to see if there was real specificity for MB

This study was set to identify as many CSF proteins as possible, by taking advantage of the very large amount of sample available from the EVD, which allowed to separate diverse fractions, to better characterize the CSF both from control and MB, in order to highlight the differently enriched proteins, and identify putative diagnostic biomarkers for the two conditions. An abundant CSF seemed promising to enable the recovery of a larger diversity of proteins. The fractions bear the potential to reveal a higher number of proteins. For example, we found that most proteins are enriched in the Exosome fraction. CPLL allowed to enrich the low abundance proteins. Each fraction displayed a list of important expressed proteins.

Then, the complex of results from all the fractions led us to select the two diagnostic biomarkers. We chose the proteins that had the top priority position in all examined fractions. For data validation we used the total in order not to manipulate the sample, avoiding those errors due to methodology. We did not test the biomarkers on each fraction because it appeared correct to study the total that encompasses all the fractions considered, therefore is omni-comprehensive; also, the material is easier to manipulate. Of course, the two chosen biomarkers were highlighted in all the fractions (see position rank), with the notable exception of CPLLs, which enrich proteins at low concentration and therefore equalize the samples.

Validation of Western blot should be performed on the 6 samples and the raw data should be presented.

Indeed, validation has been performed on the 6 + 6 samples by ELISA assay, chosen as it is very sensitive and quantitative. ELISA assay was performed on a group of 61 patients consisting of the previous 12 subjects with the addition of 18 congenital hydrocephalus unrelated to a brain tumor, 16 low-grade Gliomas and Glioneuraltumors, 6 Medulloblastoma, and 9 other brain tumor. Conversely, WB validation has only be performed for LMNB1 on the total fraction of CSF from EVD of control and MB patients; it seemed redundant to perform all of it, moreover it would not have been possible to compare the data with that from the other cancer types (see Figure 4).Also, the little time allowed by the Editors for the revision (10 days) would not allow us to perform what requested.

Validation of the other marker SLC27A$ should've been performed.

The reason we chose to validate with ELISA the LMNB1 biomarker and not SLC27A4 is because it is easier to study the significance of an increase rather than of a decrease. Actually, LMNB1 was not the first highlighted protein, it was the second one: the first one was the Long fatty acid carrier(SLC1), which however decreased, therefore we chose a protein that was increased (see Supplemental Table 3, column "rank fraction”)

the title is misleading as WGCNA is not an artificial intelligent approach but a method to value the correlation among network analysis. Pathway analysis and systems biology should have been performed

We agree with the reviewer if by AI they refer to finding algorithms that distinguish the various conditions: that was not our case.

However, we considered the machine-learning methods such as non-linear support vector machine (SVM) learning, and partial least squares discriminant analysis (PLS-DA) techniques close to AI, in fact we used it to categorize the samples and then to establish a RANK of importance of the proteins highlighted in the individual fractions (total, EV, CSF, and MV and CPLL). This was also the opinion of the Editors, who stated that our work fell under the general topic of the Special Issue (Artificial Intelligence in Cancer Metabolism and Metabolomics) to which we sent our manuscript when we made a presubmission enquiry. We changed the title of the revised version of the manuscript to Weighted gene co-expression network analysis and support vector machine learning in proteomic profiling of Cerebrospinal fluid from extraventricular drainage in child Medulloblastoma. We have also brought this issue to the attention of the Editors, as if we remove the reference to IA in the title our manuscript could no longer fall under Special Issue, and this would be a problem.

no power analysis was presented to the 6 samples and they are highly heterogeneous

Actually, power analysis was, of course, performed according to: Dell R.B., Holleran S., and Ramakrishnan R. Sample size determination. ILAR J. 2002; 43 (4):207-13. doi: 10.1093/ilar.43.4.207. only it was not cited. This was added to the revised version of the manuscript.

Reviewer 2 Report

The work entitled "Artificial Intelligence in Proteomic Profiling of Cerebrospinal Fluid from Extraventricular Drainage in Child Medulloblastoma” aims to provide a comprehensive proteomic analysis human CSF Exo,somes. This study is valuable for discovery potential targets, which resulted in the identification of LMNB1 further validated by WB. Overall the manuscript is well written, but should take in consideration:

·       A resume integrative picture could be done to provide an illustration of data and transpose it.

·       In title “Artificial Intelligence in….” should be removed. In fact authors have performed a proteomic characterization to find potential markers. AI aims to create a correlation between clinical data and proteomic data. Authors have done separated analysis where no mixture of data. It will be important develop algorithms for stratification. Why was not considered?  

·       Authors have correlated data with vesiclepedia?

·       Which advantages come with this study?

Data access by PRIDE is not available.

Author Response

Open Review-2

Comments and Suggestions for Authors

The work entitled "Artificial Intelligence in Proteomic Profiling of Cerebrospinal Fluid from Extraventricular Drainage in Child Medulloblastoma” aims to provide a comprehensive proteomic analysis human CSF Exosomes. This study is valuable for discovery potential targets, which resulted in the identification of LMNB1 further validated by WB. Overall the manuscript is well written, but should take in consideration:

  • A resume integrative picture could be done to provide an illustration of data and transpose it.

We are grateful to the Reviewer for this suggestion. Accordingly, a resume integrative schematic illustrating the data has been added to the revised version of the manuscript, as new Figure 1.

       In title “Artificial Intelligence in….” should be removed. In fact authors have performed a proteomic characterization to find potential markers. AI aims to create a correlation between clinical data and proteomic data. Authors have done separated analysis where no mixture of data. It will be important develop algorithms for stratification. Why was not considered?  

We agree with the reviewer as far as finding algorithms that distinguish the various conditions are concerned. Neither our group or IRCCS is able do this. However, we considered the machine-learning statistical assay “Support Vector Machine” a technique close to AI, in fact we used it to categorize the samples and then to establish a RANK of importance of the proteins highlighted in the individual fractions (total, EV, CSF, and MV and CPLL). Also WGCNA correlates clinical data to proteomic data (and it did not find any other correlation than those cited).

This was also the opinion of the Editors, who stated that our work fell under the general topic of the Special Issue (Artificial Intelligence in Cancer Metabolism and Metabolomics) to which we sent our manuscript when we made a presubmission enquiry. We changed the title of the revised version of the manuscript to “Weighted gene co-expression network analysis and support vector machine learning in proteomic profiling of Cerebrospinal fluid from extraventricular drainage in child Medulloblastoma”. We have also brought this issue to the attention of the Editors, as if we remove the reference to IA in the title our manuscript could no longer fall under Special Issue, and this would be a problem.

  • Authors have correlated data with vesiclepedia?

Yes. The correlation of data with vesiclepedia had been done. In Protein composition section (2.2) and Supplemental. Table 4 it is clearly stated that both Exocarta and vesiclepedia were used to identified the already described proteins and analyze the Gene Ontology (GO) enrichment.

Which advantages come with this study?

The advantage of the study was to make the most of the very large amount of “waste” CSF sample available from the EVD daily, to identify as many proteins as possible. An abundant CSF seemed promising to enable the recovery of a larger diversity of proteins; in fact, it allowed to separate diverse fractions: a more through characterization of the CSF both from controls and MB patients (regardless of their subtype), in order to highlight the differently enriched proteins, and identify putative diagnostic biomarkers for the two conditions. For example, we found that most proteins are enriched in the Exosome fraction. CPLL allowed to enrich the low abundance proteins. Each fraction displayed a list of important expressed proteins.

Then, the complex of results from all the fractions led us to select the two diagnostic biomarkers.

Data access by PRIDE is not available.

We apologize for this lack. Indeed, at the time of the submission we had applied for it, but the answer had not yet come, now the data access by PRIDE is available.

Round 2

Reviewer 1 Report

we thank the reviewers effort to answer, the work has major flaws and discrepancies,

I do not agree these can be solved by just discussion with me:

1-You cant not have 6 heterogeneous clinical samples especially it is a cancer:

It is not an advantage !

2-You can not use western on 1 sample and ELISA on other Samples:  you are not doing a school report, it is a clinical study.  and justifying saying downregulated is hard to validate !!!!!!!

Reviewer 2 Report

The paper is acceptable for publication.